# Role of the JAK2/STAT3 pathway on infection of *Francisella novicida*

**Sonoko Matsumoto**[1☯], **Takashi Shimizu**[2☯]*, **Akihiko Uda**[3], **Kenta Watanabe**[1], **Masahisa Watarai**[1]

1 Laboratory of Veterinary Public Health, Joint Faculty of Veterinary Medicine, Yamaguchi University, Yamaguchi, Japan, 2 One Welfare Education and Research Center, Joint Faculty of Veterinary Medicine, Yamaguchi University, Yamaguchi, Japan, 3 Department of Veterinary Science, National Institute of Infectious Diseases, Shinjuku, Tokyo, Japan

☯ These authors contributed equally to this work.
* shimizut@yamaguchi-u.ac.jp

**Data Availability Statement:** All relevant data are within the paper and its Supporting Information files.

**Funding:** "SM: JST SPRING Grant Number JPMJSP2111, TS: JSPS KAKENHI Grant Number

## Abstract

*Francisella tularensis* is a causative agent of the zoonotic disease tularemia, and is highly pathogenic to humans. The pathogenicity of this bacterium is largely attributed to intracellular growth in host cells. Although several bacterial factors important for the intracellular growth have been elucidated, including the type VI secretion system, the host factors involved in the intracellular growth of *F. tularensis* are largely unknown. To identify the host factors important for *F. tularensis* infection, 368 compounds were screened for the negative regulation of *F. tularensis* subsp. *novicida* (*F. novicida*) infection. Consequently, 56 inhibitors were isolated that decreased *F. novicida* infection. Among those inhibitors, we focused on cucurbitacin I, an inhibitor of the JAK2/ STAT3 pathway. Cucurbitacin I and another JAK2/STAT3 inhibitor, Stattic, decreased the intracellular bacterial number of *F. novicida*. However, these inhibitors failed to affect the cell attachment or the intrasaccular proliferation of *F. novicida*. In addition, treatment with these inhibitors destabilized actin filaments. These results suggest that the JAK2/STAT3 pathway plays an important role in internalization of *F. novicida* into host cells through mechanisms involving actin dynamics, such as phagocytosis.

## Introduction

*Francisella tularensis* are gram-negative bacteria that can be distinguished into four subspecies; *F. tularensis* subsp. *tularensis* (type A), *F. tularensis* subsp. *holarctica* (type B), *F. tularensis* subsp. *mediasiatica*, and *F. tularensis* subsp. *novicida* (*F. novicida*). Among them type A and type B *F. tularensis* have been reported to be pathogenic to humans [1]. *F. tularensis* is the zoonotic causative agent of tularemia, which occurs in Northern Hemisphere countries [1]. As few as 10 bacteria of type A *F. tularensis* inhaled through aerosols develop tularemia in humans [2]. Consequently, because of the high infectivity of *F. tularensis*, the Centers for Disease Control and Prevention (USA) is concerned for misuse of *F. tularensis* in terrorism and has classified this bacterium as a category A bioterrorism agent [3]. *F. novicida* is highly related to type

22K07054, MW: JSPS KAKENHI Grant Number 21H02360" The funders had no role in study design, data collection and analysis, decision to publish, or preparation of the manuscript.

**Competing interests:** The authors have declared that no competing interests exist.

A *F. tularensis* genetically but exhibits low pathogenicity in humans [1]. However, *F. novicida* is pathogenic to mice and is a commensal intracellular pathogen that replicates in both human and mouse macrophages in the same way as seen with type A *F. tularensis* [4]. Thus *F. novicida* is thought to have considerable homology with type A *F. tularensis* and is often used as a practical surrogate of type A *F. tularensis* [5]. Although there are many reports on virulence factors of *Francisella*, including the presence of a type VI secretion system [6], the host factors important for the infection of the bacteria are largely unknown.

*Francisella* are ingested through the pseudopodial loop of macrophages and incorporated into spacious vacuoles that have endosomal markers [7, 8]. The bacterium then escapes from the phagosomal membrane and replicates in the cytoplasm [9]. In the late stages of infection, *Francisella* bacteria re-enter the autophagosome [10] and acquire amino acids from degraded proteins and replicate [10]. Cytoplasmic bacteria with defective or damaged replication are trapped in *Francisella*-containing vacuoles, which are lysosome-associated membrane protein 1 (LAMP-1)-positive autophagosomes and are degraded via the ubiquitin-SQSTM1-LC3 pathway [11, 12].

The JAK2/STAT3 pathway is involved in various biological processes such as immunity, cell division, cell death, and tumor formation [13]. Activation of the JAK2/STAT3 pathway occurs via the binding of the extracellular domain of specific intracellular receptors (RTKs) to hormones (e.g., prolactin), growth factors (e.g., epidermal growth factor, EGF), and cytokines (e.g., the interleukin-6, IL-6, family). JAK2 mediates signaling through several cytokine receptors, including IL-6 and IFN-γ. The interaction between ligands and receptors induces dimerization of the receptor subunits. The close proximity of JAK2 noncovalently bound to the intracellular domain of the receptor causes autophosphorylation of JAK2 and stimulates the kinase activity [14, 15]. When the JAK2 protein is phosphorylated, tyrosine residues in the intracellular domain of the receptor are phosphorylated by the activated JAK2 kinase domain, creating a docking site for STAT3 within the SH2 domains of the receptor. This allows cytoplasmic STAT3 protein to bind to the phosphorylated tyrosine residues on the receptor. STAT3 is then phosphorylated by JAK2, dimerizes, dissociates from the receptor, and moves to the nucleus. The phosphorylated STAT3 dimer binds to specific DNA sequences, inducing transcription of target genes, including Cyclin D1, cMyc, Bclx1, bcl-2, MCL-1, and P53, leading to cell proliferation, differentiation, apoptosis, and immune regulation [15]. Furthermore, JAK2/STAT3 signaling can interact with other pathways, including the MAPK/ERK and PI3K/ACT/mTOR signaling pathways, to activate specific cellular responses [16].

In this study, we sought to identify the host factors important for *Francisella* infection and screened 368 compounds for those that inhibited *F. novicida* infection. Consequently, we focused on cucurbitacin I, an inhibitor of JAK2/STAT3 [17], and investigated the effect of JAK2/STAT3 pathway on *F. novicida* infection.

## Materials and methods

All experiments were conducted in accordance with the institutional biosecurity guidelines and were approved by Yamaguchi University.

### Inhibitor library and inhibitors

The inhibitor compound library was obtained from Molecular Profiling Committee, Grant-in-Aid for Transformative Research Areas "Advanced Animal Model Support (AdAMS)" from The Ministry of Education, Culture, Sports, Science, and Technology, Japan (JSPS KAKENHI Grant Number JP 22H04922). Cucurbitacin I (Merck, Darmstadt, Germany) and Stattic (Merck) were dissolved in dimethyl sulfoxide (DMSO) at 2 mM and then diluted to 200, 20,

and 2 μM. The same amount of the inhibitors and DMSO control were added to culture medium at a final concentration of 10, 1, 0.1, and 0.01 μM.

## Cell culture

The mouse monocyte-macrophage J774.1 cell line was cultured at 37°C under 5% $CO_2$ in Roswell Park Memorial Institute (RPMI) 1640 medium (Thermo Fisher, Waltham, MA) supplemented with 10% fetal bovine serum (Thermo Fisher).

## Bacteria strains and culture conditions

*F. novicida* ATCC 15482 strain was cultured aerobically at 37°C with brain heart infusion broth (Becton, Dickinson and Company, Franklin Lakes, NJ) supplemented with cysteine (BHIc), BHIc plates containing 1.5% Agar (Wako Laboratory Chemicals, Osaka, Japan) [18], or chemically defined medium (CDM) [19]. Green-Fluorescent protein (GFP)-expressing *F. novicida* was cultured with BHIc containing 2.5 μg/mL chloramphenicol [20]. *Escherichia coli* ATCC15482 strain was cultured aerobically at 37°C in Luria–Bertani medium (LB) (Nacalai Tesque, Kyoto, Japan) or LB plates containing 1.5% agar. Bacterial concentrations were adjusted based on their optical density at 595 nm.

## Screening of inhibitors

J774.1 cells ($25 \times 10^4$ cells/mL) were seeded at 100 μL/well in a 96-well plate or 500 μL/well in a 24-well plate and cultured overnight. Cells were treated with 10 or 1 μM of inhibitors for 2 h; DMSO was used as the control. After treatment, cells were infected with GFP-expressing *F. novicida* at multiplicity of infection (MOI) of 1 and incubated for 24 h. Cells were then washed three times with phosphate-buffered saline (PBS) and fluorescence intensity was measured using plate reader 2030 ARVO X4 (Perlin Elmer, Waltham, MA). An intensity at 4,000 lower than that of the DMSO control showed inhibition of infection, whereas an intensity at 5000 higher than that of the control showed enhancement of infection (S1 Table).

## Inhibitor treatment before infection

J774.1 cells ($25 \times 10^4$ cells/mL) were seeded at 100 μL/well in a 96-well plate or 500 μL/well in a 24-well plate and cultured overnight. Cells were treated with indicated concentration of cucurbitacin I or Stattic for 2 h; DMSO was used as the control. After treatment, cells were infected with *F. novicida* at an MOI of 1. Plates were centrifuged for 10 min at $300 \times g$ and incubated for 1 h at 37°C. Cells were cultured in culture medium containing inhibitors, and 50 μg/mL of gentamycin added for 1 h to kill extracellular bacteria. Cells were then washed three times with PBS and cultured in medium containing inhibitors at 37°C.

## Inhibitor treatment after infection

J774.1 cells ($25 \times 10^4$ cells/mL) were seeded at 100 μL/well in a 96-well plate or 500 μL/well in a 24-well plate and cultured overnight. Cells were infected with *F. novicida* at an MOI of 1. Plates were centrifuged for 10 min at $300 \times g$ and incubated for 1 h at 37°C. Cells were treated with 50 μg/mL of gentamycin to kill extracellular bacteria. Cells were washed three times with culture medium and treated with indicated concentrations of cucurbitacin I or Stattic.

## Colony forming units (CFU)

J774.1 cells ($25 \times 10^4$ cells/mL) were seeded at 100 μL/well in a 96-well plate and cultured overnight. After inhibitor treatment and infection described above, cells were washed three times

with PB and then disrupted with 0.1% Triton-X in CDM for 1 min and 900 μL of CDM was immediately added. Samples were diluted with CDM and cultured on a BHIc plate overnight, and the number of colonies were counted.

## Laser scanning confocal microscopy

J774.1 cells ($25 \times 10^4$ cells/mL) were seeded at 500 μL/well in a 24-well plate with 120-mm glass coverslips (Matsunami, Osaka, Japan) and cultured overnight. After treatment with inhibitors and infection with GFP-expressing *F. novicida* as described above, cells were washed three times with PBS and fixed with 4% paraformaldehyde in PBS for 30 min at room temperature. Images of the cells were obtained using FluoView FV100 confocal laser scanning microscope (Olympus, Tokyo, Japan).

## Measurement of phagocytotic activity

The activity of phagocytosis against *E. coli* was measured as previously described with slight modification [21, 22]. J774.1 cells ($25 \times 10^4$ cells/mL) were seeded at 100 μL/well in a 96-well plate or 500 μL/well in a 24-well plate and cultured overnight. Cells were treated with the indicated concentration of cucurbitacin I, Stattic, or DMSO control for 2 h. After treatment, cells were infected with *E. coli* for 3 h and treated with gentamycin to kill extracellular bacteria. Cells were then washed three times with PBS and disrupted with 0.1% Triton-X in PBS for 1 min followed by immediate addition of 900 μL PBS. CFU was counted as described above.

## Visualization of actin filaments

J774.1 cells ($25 \times 10^4$ cells/mL) were seeded at 500 μL/well in a 24-well plate with 12-mm glass coverslips and cultured overnight. After treatment of inhibitors and infection described above, cells were washed three times with PBS and fixed with 4% paraformaldehyde in PBS. After being washed three times with PBS, cells were permeabilized with 0.1% Triton-X in PBS for 5 min and washed three times with PBS. Cells were then blocked with 2% of bovine serum albumin in PBS for 1 h. and stained with 0.1 μM of phalloidin-rhodamine B isothiocyanate (P1951, Thermo Fisher) for 2 h. Cells were washed three times with PBS and observed with laser scanning confocal microscopy.

## Statistical analysis

Significant differences are determined by $P < 0.05$ or $P < 0.01$ using Student's t-test or Dunnett's test, or Tukey–Kramer method, and indicated with * and ** respectively.

# Results

## Screening of inhibitors

To identify host factors important for the infection of *F. novicida*, 368 inhibitor compounds were screened to identify those that inhibited the growth of *F. novicida*. J774.1 cells treated with inhibitors for 2 h were infected with GFP-expressing *F. novicida*, and cells with a lower fluorescence compared with that of untreated cells were selected (**S1 Table**). Consequently, 56 inhibitors were selected to inhibit the growth of *F. novicida* in J774.1 cells (Table 1). A number of the identified inhibitors have been demonstrated to possess antibiotic properties. Among the non-antibiotic inhibitors, three were found to be involved in the Jak-2/Stat3 pathway. Consequently, we considered that the Jak-2/Stat3 pathway was a key factor in *F. novicida* infection and focused on cucurbitacin I as a representative inhibitor of the JAK2/STAT3 pathway.

**Table 1. Inhibitors negatively regulated the infection of *F. novicida*.**

| Compound | Category |
| --- | --- |
| 5,15-DPP | STAT3 |
| ABT-702 | AK |
| Actinonin | aminopeptidase M |
| AG957 | Bcr-abl |
| Amastatin | aminopeptidase A |
| AMD3100 octahydrochloride | CXCR4 |
| Aminoglutethimide | aromatase |
| anisomycin | stress inducer |
| Bafilomycin A1 | V-ATPase |
| BH3I-1 | Bcl-XL |
| bortezomib | Proteasome |
| brefeldin A | golgi inhibitor |
| CA-074 | cathepsin B |
| Cdk4 inhibitor | CDK |
| Chetomin | HIF |
| Cucurbitacin I | Jak-2 |
| cyclopamine | Hedgehog |
| Cytochalasin D | actin filament |
| E-64d | calpain |
| Formestane | aromatase |
| gefitinib | EGFR |
| Glibenclamide | K channel |
| HA 14–1 | Bcl-2 |
| imatinib mesylate | Bcr-Abl/Kit |
| Jervine | Hedgehog |
| KT5823 | PKG |
| Leptomycin B | CRM1 |
| LFM-A13 | Burton's tyrosine kinase(BTK) |
| MG-132 | proteasome |
| Mifepristone | progesterone receptor |
| Monensin | Na ionophore |
| Nifedipine | Ca channel |
| Nigericin | K ionophore |
| nilotinib | Bcr-Abl |
| NSC625987 | CDK |
| NSC95397 | Cdc25 |
| Ouabain | Na/K ATPase |
| Pepstatin A | cathepsin D |
| Pifithrin-a (cyclic) | p53 |
| PIM1 Inhibitor II | PIM |
| PKR inhibitor | PKR |
| PRIMA-1 | p53 activator |
| radicicol | Hsp90 |
| Rotenone | mitochondrial complex I |
| RS 102895 | CCR2 |
| Sanguinarine | Na/K/Mg ATPase |
| SB 328437 | CCR3 |

(*Continued*)

**Table 1.** (Continued)

| Compound | Category |
|---|---|
| sorafenib | Multi-kinases |
| sunitinib malate | Multi-kinases |
| temsirolimus | mTOR |
| Thapsigargin | Ca-ATPase |
| TOFA | acetyl-CoA carboxylase (ACC) |
| vorinostat | HDAC |
| WP1066 | STAT3 |
| YM155 | Survivin |
| Z-GLF-CMK | cathepsin G |

## Effect of inhibitors on *F. novicida* infection

To assess whether cucurbitacin I suppressed *F. novicida* infection, J774.1 cells treated with cucurbitacin I were infected them with GFP-expressing *F. novicida*, and intracellular bacteria were observed using confocal microscopy. Cucurbitacin I-treated J774.1 cells were also infected with *F. novicida*, and the number of intracellular bacteria was measured by colony counting. Cucurbitacin I treatment decreased the number of intracellular *F. novicida* (Fig 1A, 1C), indicating that the JAK2/STAT3 pathway is important for *F. novicida* infection. To confirm this, the effect of another JAK2/STAT3 pathway inhibitor Stattic was evaluated. Stattic treatment decreased the number of intracellular *F. novicida* as assessed by both microscopic observation and colony counting (Fig 1B, 1D). A concentration of 1 μM cucurbitacin and 10 μM of Stattic were significantly effective and were therefore used in subsequent experiments.

## Growth of *F. novicida* in BHIc

To check the direct effect of inhibitors on the growth of *F. novicida*, cucurbitacin I or Stattic was added into the growth medium, and the growth was measured by optical density and colony count. No significant differences in counts were observed between inhibitors and DMSO control (Fig 2A–2D). GFP-expressing *F. novicida* cultured with inhibitors were washed and infected into J774.1 cells, and intracellular bacteria were observed using confocal microscopy. The same levels of intracellular *F. novicida* was observed in bacteria cultured with cucurbitacin I or Stattic compared with that of DMSO control (Fig 3). Therefore, cucurbitacin I and Stattic had no direct effect on the growth and infectivity of *F. novicida*.

## Cell adhesion, invasion, and intracellular proliferation of *F. novicida*

Infection of *F. novicida* is established via three infections steps: attachment, internalization, and proliferation. To investigate which step of infection is affected by the identified inhibitors, the attachment of *F. novicida* was initially tested. J774.1 cells were treated with cucurbitacin or Stattic and then infected with *F. novicida*. The number of bacteria attached to cells just after infection (10 and 30 min) was measured by colony counting. No significant difference was observed in colony counts between control DMSO and inhibitor treatments (Fig 4). Next, to test the effect of inhibitors of internalization and proliferation, J774.1 cells were treated with inhibitors and infected with *F. novicida*. Cells were incubated for 1 h to allow internalization of bacteria while attached bacterial cells were removed by gentamicin treatment. After 1.5 and 12 h incubation, the number of internalized and proliferated *F. novicida* were measured. The number of bacteria was decreased in cells treated with each inhibitor compared with that

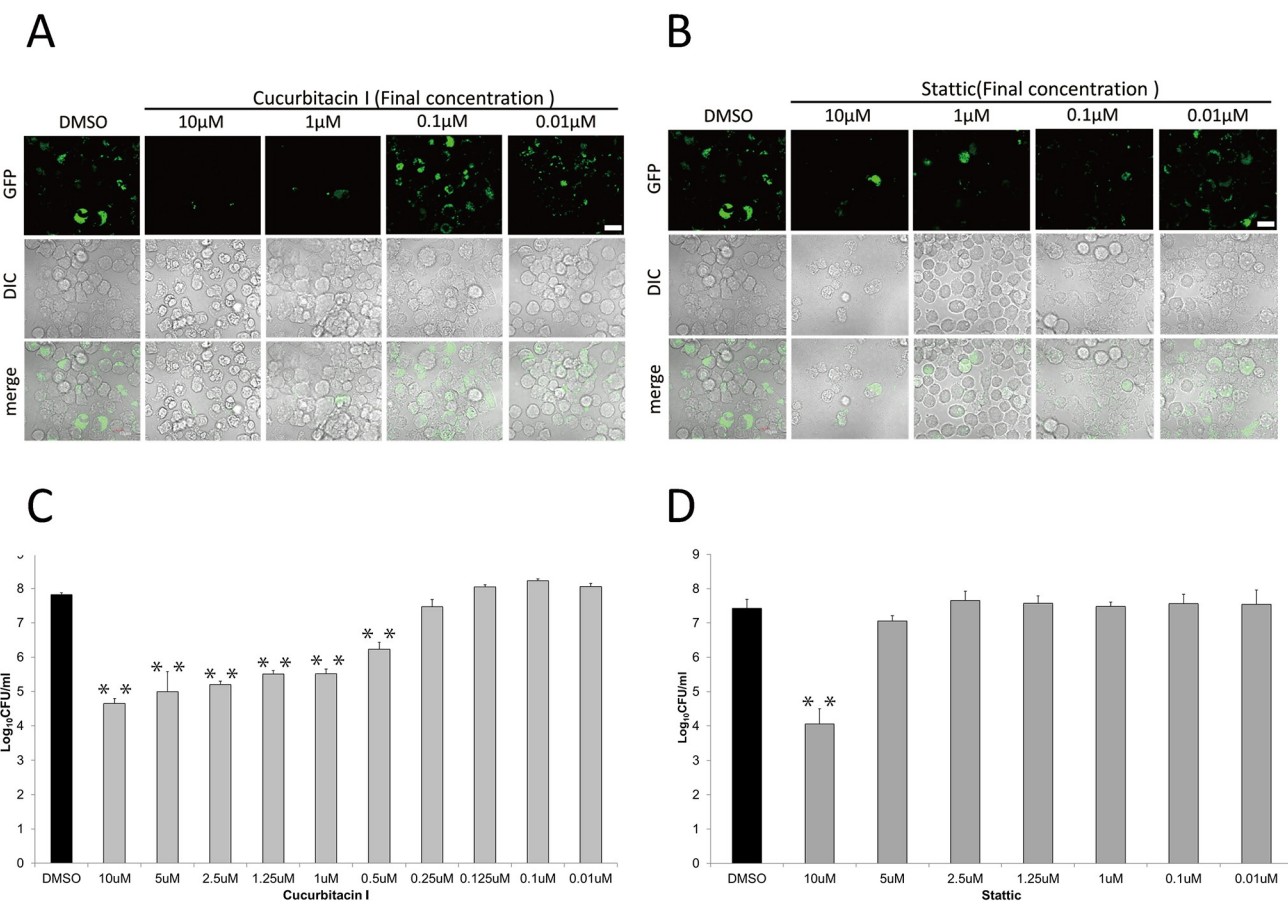

**Fig 1. Suppression of *F. novicida* infection by JAK2/STAT3 inhibitors.** J774.1 cells treated with 0.01 to 10 µM of cucurbitacin I (A, C) or Stattic (B, D) were infected with *F. novicida* (C, D) or GFP-expressing F. novicida (A, B). Cells were treated with gentamicin and incubated for 12 h and then observed with confocal microscopy (A, B), or the number of intracellular bacterial was counted (C, D). Data represent the average and standard deviation of three identical experiments. Differences were analyzed with multiple comparison (Dunnett's test) and indicated by asterisks, **$P < 0.01$, *$P < 0.05$. Scale bar: 10µm.

treated with DMSO, indicating that either internalization or proliferation were affected by the inhibitors (Fig 5). To determine which steps of internalization and proliferation was affected by inhibitors, the proliferation of *F. novicida* was examined. J774.1 cells were infected with *F. novicida* and incubated for 1 h to allow internalization of bacteria. Attached bacteria were removed by gentamicin treatment and infected cells were treated with inhibitors. The number of bacteria in inhibitor-treated cells were unaffected by inhibitors (Fig 6). Thus, cucurbitacin I and Stattic affect internalization but not proliferation in *F. novicida* infection.

## Phagocytotic activity

As cucurbitacin I and Stattic affected the internalization of *F. novicida*, the effect of inhibitors on phagocytosis was investigated using the nonintracellular bacteria *E. coli*. J774.1 cells were treated with cucurbitacin I or Stattic and infected with *E. coli*. Cells were incubated for 3 h to allow phagocytosis and attached cells were removed by gentamicin treatment for 30 min. At 3.5 h post infection, the internal bacterial number was measured by colony counting. Addition of inhibitors decreased the number of internalized *E. coli* (Fig 7), indicating cucurbitacin I and Stattic affect host cell phagocytosis.

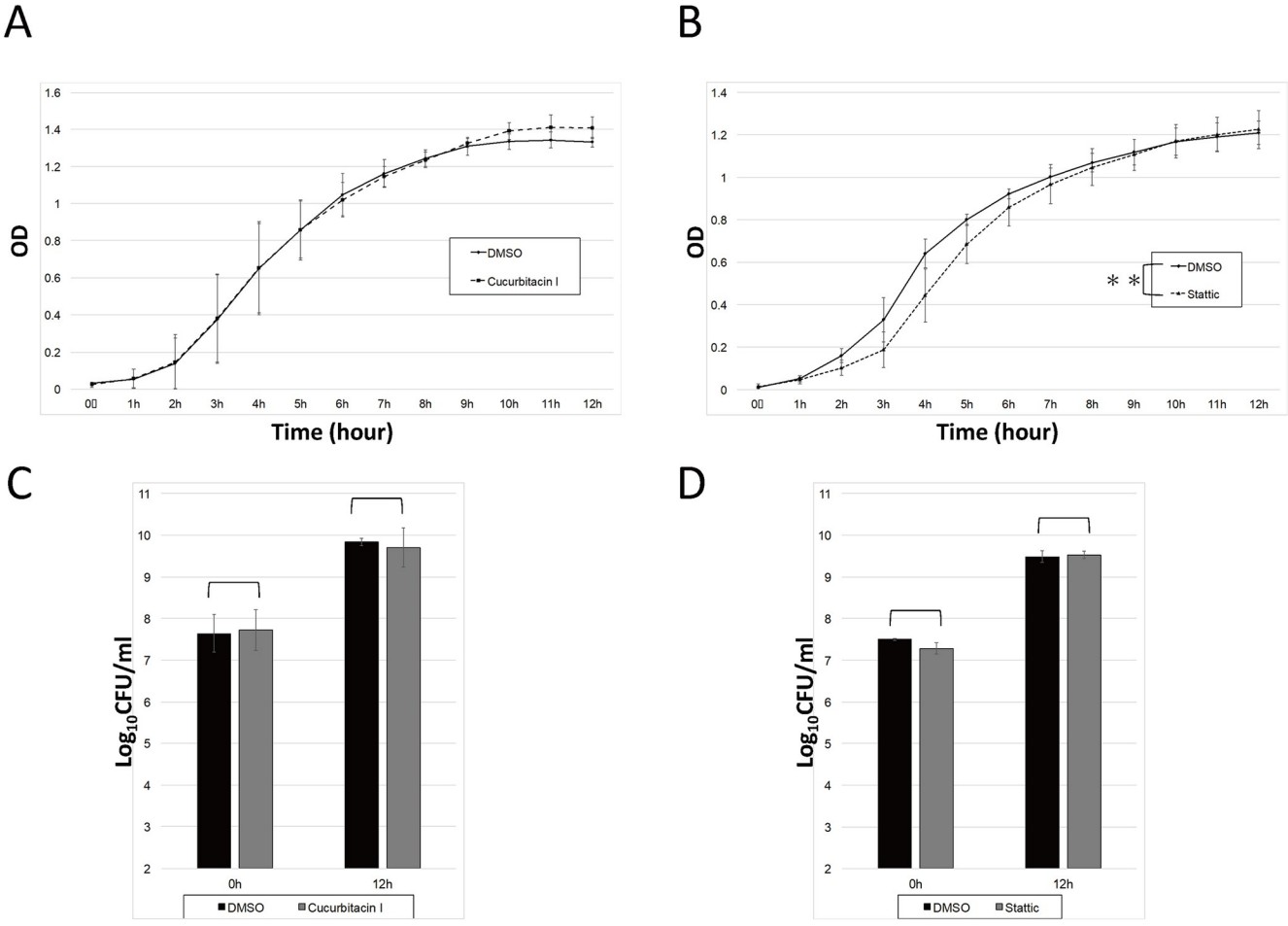

**Fig 2. Growth of *F. novicida* in culture medium with inhibitors.** *F. novicida* was cultured in BHIc with cucurbitacin I (A, C) or Stattic (B, D) and optical density (λ = 595 nm) was measured at the indicated time point (A, B). The number of CFU at 0 and 12 h was counted (C, D). Data represent the average and standard deviation of three identical experiments. Differences were analyzed with multiple comparison (Tukey–Kramer method) (A, C) or Student t-test (B, D) and indicated by asterisks, *$P < 0.05$.

## Actin filaments

Phagocytosis results from polymerization, depolymerization, and rearrangement of actin [23]. To investigate the effect of inhibitors on actin polymerization, J774.1 cells were treated with cucurbitacin I or Stattic for 2 h and infected with *F. novicida*, and then actin was visualized using immunofluorescence microscopy. Abnormal arrangements of actin were observed in inhibitor-treated cells compared with those in cells treated with DMSO control (Fig 8). These results suggest that the correct arrangement of actin is crucial for the infection of *F. novicida*.

## Discussion

To identify the host factors important for *Francisella* infection, 368 inhibitors were screened, and those that affected *F. novicida* infection were selected. Consequently, 56 compounds inhibited the infection of *F. novicida* while eight enhanced the infection. In this study we focused on 56 inhibitors that negatively affected infection to identify the host factors important for infection. Most of the 56 inhibitors possessed antibiotic property, whereas three inhibitors were related to the JAK2/STAT3 pathway. Therefore, we focused on the inhibitors related to

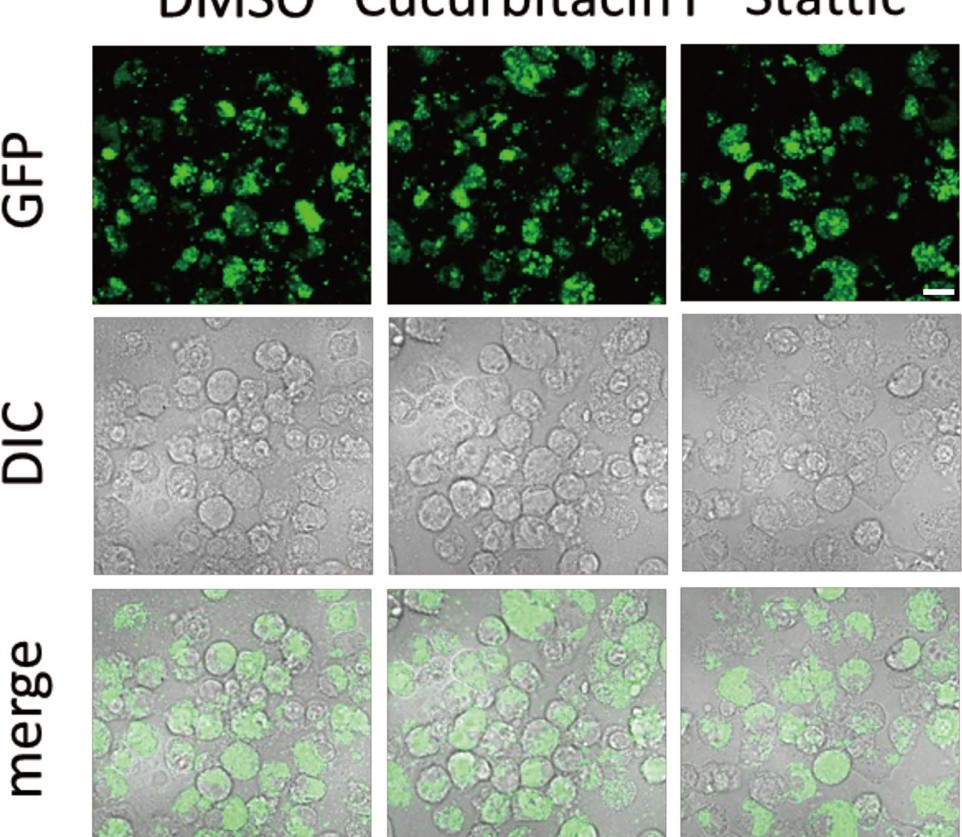

**Fig 3. Intracellular growth of *F. novicida* cultured with inhibitors.** *F. novicida* was cultured in BHIc with cucurbitacin I or Stattic. J774.1 cells were infected with inhibitor-treated *F. novicida* and observed at 12 h post infection. Scale bar: 10μm.

the JAK2/STAT3 pathway without antibiotic properties, and cucurbitacin I was selected for farther study.

Cucurbitacin I is a triterpenoid compound derived from the fruit extract of plants, such as cucumber, in the *Cucurbitaceae* family [24]. Cucurbitacin I inhibits JAK2 phosphorylation and thereby suppresses the levels of tyrosine-phosphorylated STAT3 [17]. In this study, cucurbitacin I inhibited the infection of *F. novicida* but failed to affect the growth of *F. novicida* in culture medium. To confirm the involvement of the JAK2/STAT3 pathway in *F. novicida* infection, another inhibitor of the JAK2/STAT3 pathway, Stattic was tested. Stattic is a non-peptide small molecule that inhibits the dimerization of STAT3 through the SH2 domain [25]. Similar to cucurbitacin I, Stattic did not affect growth in the culture medium but did inhibit *F. novicida* infection. In addition, *F. novicida* infection tended to enhance activation of STAT3 just after infection. These results indicate that the JAK2/STAT3 pathway plays an important role in *F. novicida* infection.

To examine which of the three steps of adhesion, invasion, and intracellular proliferation is inhibited by cucurbitacin I or Stattic, cells were treated with inhibitors and the subsequent effects observed at different time points. Treatment of inhibitors failed to affect the attachment of *F. novicida* to the cells at 10 or 30 min post infection, indicating that cucurbitacin I and Stattic affects the internalization or intracellular proliferation. Cucurbitacin I or Stattic treatment

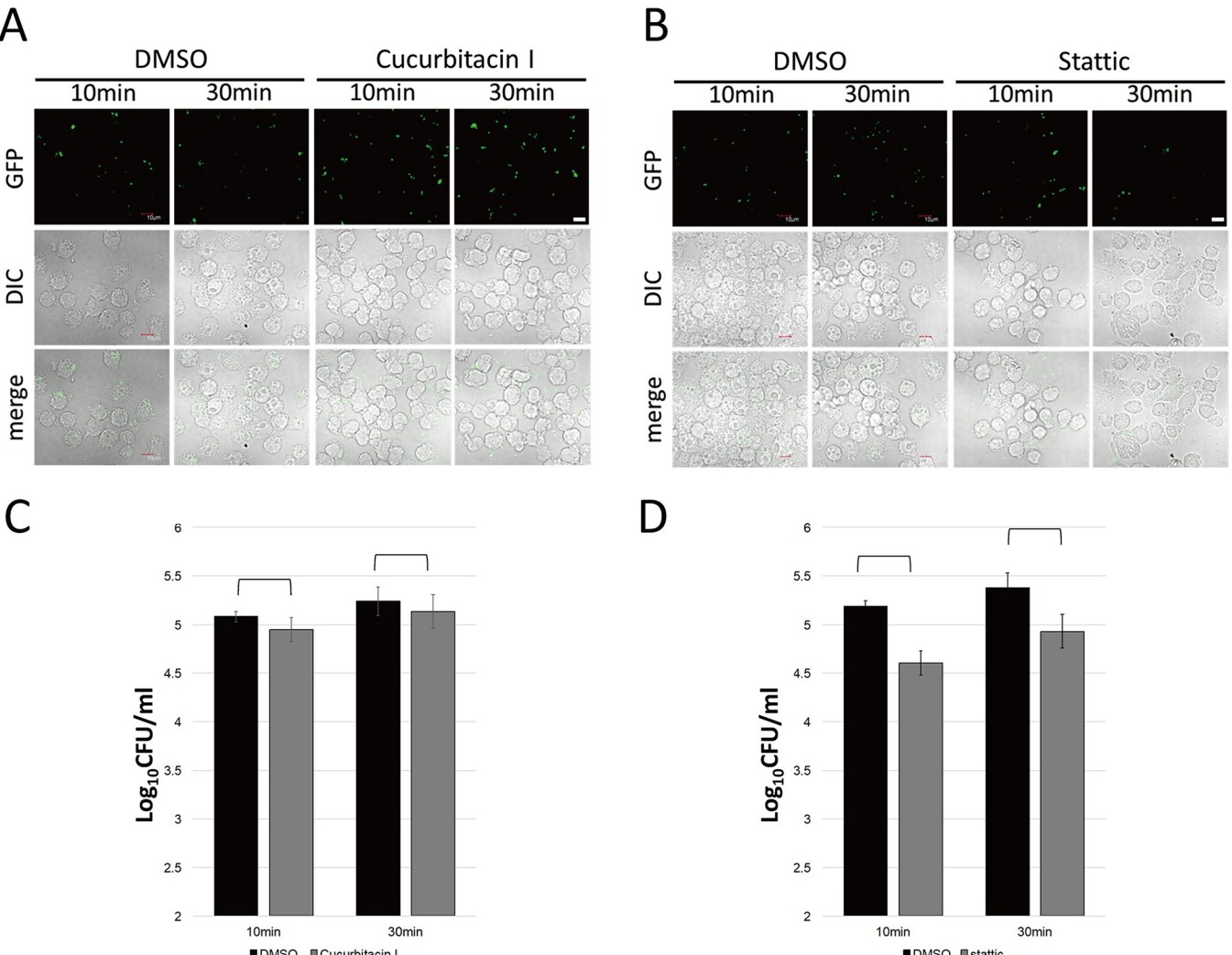

**Fig 4. Attachment of *F. novicida* to cell surface.** J774.1 cells treated with 1 μM of cucurbitacin I (A, C) or 10 μM Stattic (B, D) were infected with *F. novicida* (C, D) or GFP-expressing *F. novicida* (A, B). Cells were observed with confocal microscopy (A, B), or the intracellular bacterial number was counted (C, D) at 10 or 30 min post infection. The data represent the averages and standard deviations of three identical experiments. Differences were analyzed with Student t-test and indicated by asterisks, *$P < 0.05$. Scale bar: 10μm.

after infection failed to decrease the number of intracellular *F. novicida*, indicating that intracellular proliferation was not affected by the inhibitors, indicating that the JAK2/STAT3 pathway is important for the internalization step of *F. novicida*. These results are consistent with the various reports concerning intracellular infection by other bacteria. In infection by *Brucella abortus*, the AK2/STAT3 pathway is important for the intracellular survival of the bacteria [26]. In *Helicobacter pylori* infection, inhibition of the JAK2/STAT3 pathway reduces the development of gastric cancer [27]. In addition, the JAK2/STAT3 pathway is important for the development of pulmonary fibrosis in *Mycobacterium tuberculosis* infection [28]. Thus, the JAK2/STAT3 pathway is important for the infection and pathogenesis of various bacterial intracellular infections.

*Francisella* are ingested through the pseudopodia of macrophages and incorporated into spacious vacuoles with endosomal markers [7, 8]. The organism then escapes from the phagosomal membrane and replicates in the cytoplasm [9]. To examine which step of phagocytosis

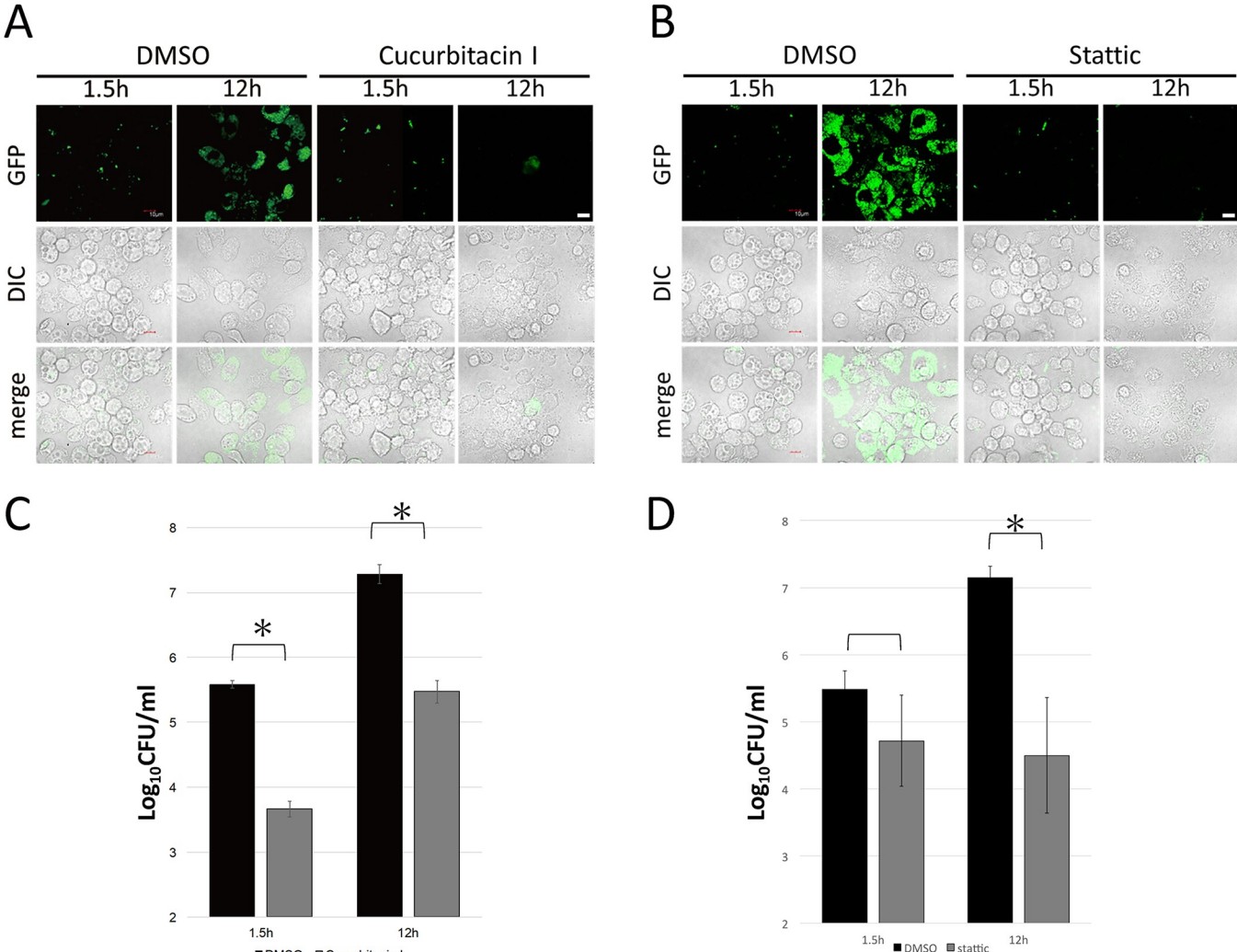

**Fig 5. Internalization and proliferation of *F. novicida*.** J774.1 cells treated with 1 μM of cucurbitacin I (A, C) or 10 μM of Stattic (B, D) were infected with *F. novicida* (C, D) or GFP-expressing *F. novicida* (A, B). Cells were treated with gentamicin and incubated for 1.5 or 12 h, Cells were then observed with confocal microscopy (A, B), or the intracellular bacterial number was counted (C, D). Data represent the average and standard deviation of three identical experiments. Differences were analyzed with Student t-test and indicated by asterisks, $^{**}P < 0.01$, $^{*}P < 0.05$. Scale bar: 10μm.

or escape from phagosome is the target of inhibitors, the ingestion of *E. coli*, a bacterium that cannot escape from the phagosome was evaluated. Subsequently, the number of ingested intracellular *E. coli* was also decreased by treatment with cucurbitacin I or Stattic. This result indicates that cucurbitacin I and Stattic inhibit the phagocytosis step of bacterial infection.

Phagocytosis results from polymerization, depolymerization, and rearrangement of actin [23], and we therefore evaluated the actin dynamics of *F. novicida*-infected cells and observed abnormal arrangements of actin in cucurbitacin I- or Stattic-treated cells. These results suggest that the JAK2/STAT3 pathway regulates actin dynamics followed by phagocytosis. This finding is consistent with a previous study where cucurbitacin I inhibited cell motility or proliferation of cancer cells by interfering with actin dynamics [29, 30].

Since cucurbitacin I exhibits an antitumor effect, cucurbitacin I and JAK2/STAT3 inhibitors have received increasing attention as potential cancer therapeutic agents [31, 32]. In this study, we identified cucurbitacin I as an inhibitor of *F. novicida* infection and demonstrated

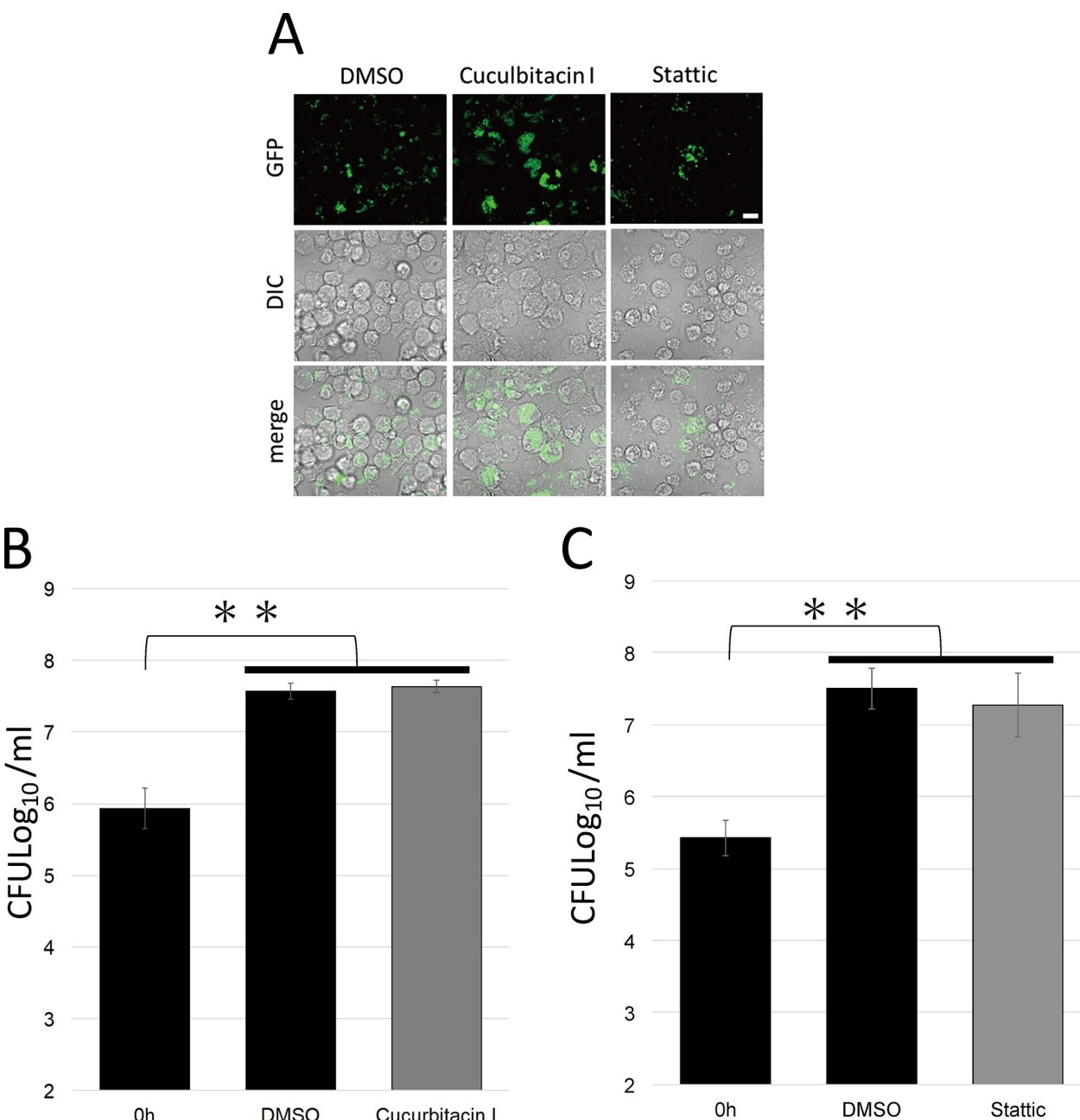

**Fig 6. Intracellular proliferation of *F. novicida*.** J774.1 cells were infected with *F. novicida* (B, C) or GFP-expressing *F. novicida* (A). Cells were treated with gentamicin and cultured with 1 μM of cucurbitacin I (A, B) or 10 μM of Stattic (A, C) for 12 h. Cells were observed with confocal microscopy(A), or the intracellular bacterial number was counted (B, C). Data represent the average and standard deviation of three identical experiments. Differences were analyzed with multiple comparison (Dunnett's test) and indicated by asterisks, **P < 0.01. Scale bar: 10μm.

that the JAK2/STAT3 pathway is important for the actin dynamics that underlie phagocytosis. In infection by other intracellular bacteria such as *Brucella* and *Mycobacterium*, the JAK2/STAT3 pathway is important for the intracellular growth and pathogenesis [26, 28]. Moreover, cucurbitacin I exhibits an antimicrobial effect through induction of autophagy [33]. Therefore, inhibitors such as cucurbitacin I and Stattic can be utilized as antimicrobial agents, and the JAK2/STAT3 pathway can be a therapeutic target of infection with intracellular bacteria as well.

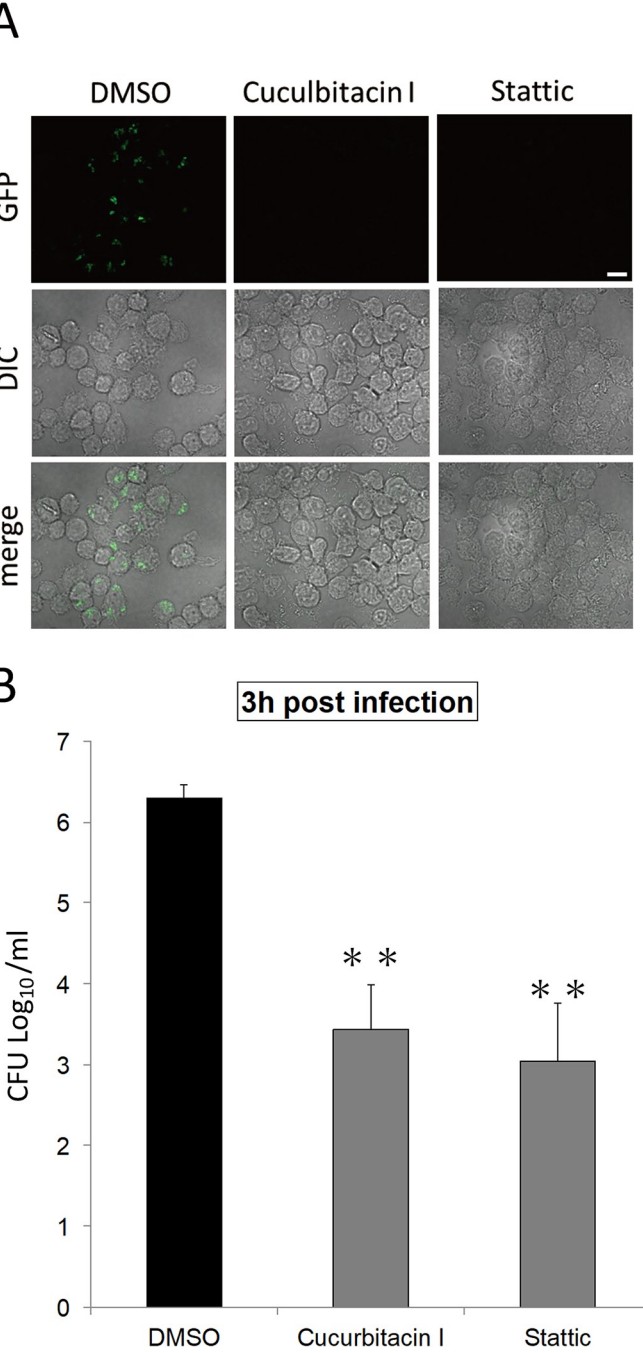

**Fig 7. Effects of inhibitors on phagocytosis.** J774.1 cells treated with 1 μM of cucurbitacin I (A, B) or 10 μM of Stattic (A, B) were infected with *E. coli* (B) or GFP-expressing *E.coli* (A). Cells were treated with gentamicin and incubated for 3 h. Cells were observed with confocal microscopy (A), or the intracellular bacterial number was counted (B). Data represent the average and standard deviation of three identical experiments. Differences were analyzed with multiple comparison (Dunnett's test) and indicated by asterisks, **$P < 0.01$. Scale bar: 10μm.

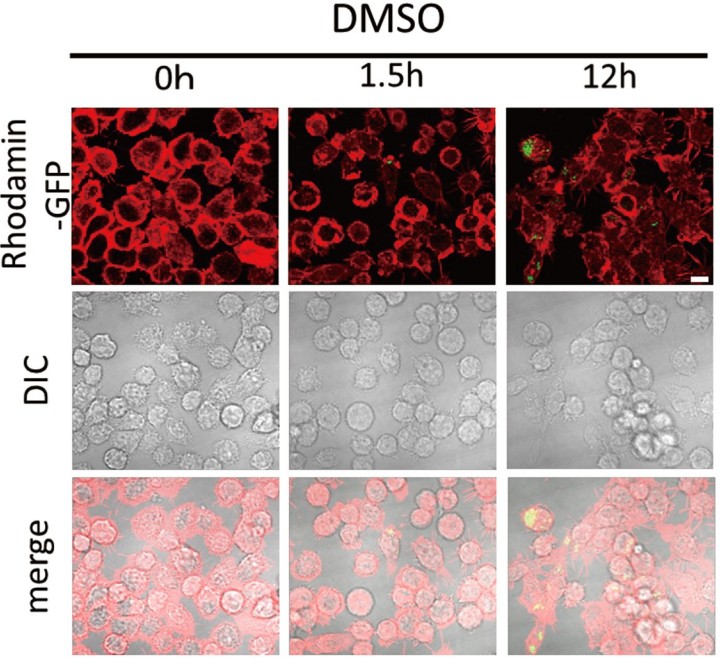

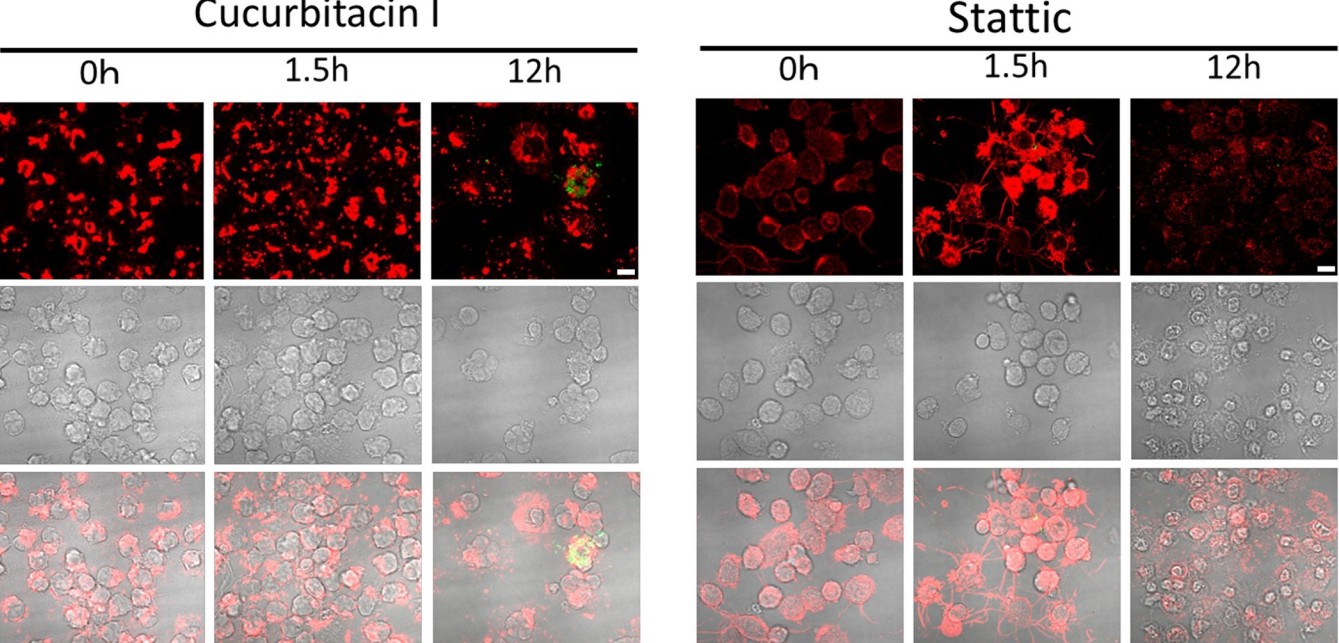

**Fig 8. Effects of inhibitors on actin filaments.** J774.1 cells treated with 1 μM of cucurbitacin I or 10 μM of Stattic were infected with GFP-expressing *F. novicida* for the indicated time. Cells were stained with phalloidin-rhodamine and observed by confocal microscopy. Scale bar: 10μm.

## Supporting information

**S1 Table. List of inhibitors.** *Fluorescence intensity of GFP-expressing *F. novicida* >5000 higher than that of the DMSO control was determined as positively regulating inhibitors (+), and >4000 lower than that of the control was determined as negatively regulating inhibitors (−). **Same compounds but derived from different providers.
(PDF)

**S1 Data.**
(XLSX)

## Author Contributions

**Conceptualization:** Sonoko Matsumoto, Takashi Shimizu, Masahisa Watarai.

**Data curation:** Sonoko Matsumoto, Takashi Shimizu.

**Formal analysis:** Sonoko Matsumoto, Takashi Shimizu, Kenta Watanabe, Masahisa Watarai.

**Funding acquisition:** Sonoko Matsumoto, Takashi Shimizu, Masahisa Watarai.

**Investigation:** Sonoko Matsumoto, Takashi Shimizu, Kenta Watanabe.

**Methodology:** Sonoko Matsumoto, Takashi Shimizu.

**Project administration:** Takashi Shimizu, Masahisa Watarai.

**Resources:** Takashi Shimizu, Akihiko Uda, Masahisa Watarai.

**Supervision:** Takashi Shimizu.

**Validation:** Takashi Shimizu.

**Visualization:** Sonoko Matsumoto, Takashi Shimizu.

**Writing – original draft:** Sonoko Matsumoto, Takashi Shimizu.

**Writing – review & editing:** Takashi Shimizu, Kenta Watanabe, Masahisa Watarai.

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
