## [Decision Letter · Decision Letter 0]

19 Aug 2024

PONE-D-24-27526Effect of the JAK2/STAT3 Pathway on Internalization of Francisella novicida.PLOS ONE

Dear Dr. Shimizu,

Thank you for submitting your manuscript to PLOS ONE. After careful consideration, we feel that it has merit but does not fully meet PLOS ONE’s publication criteria as it currently stands. Therefore, we invite you to submit a revised version of the manuscript that addresses the points raised during the review process.

We look forward to receiving your revised manuscript.

Kind regards,

Ebrahim Shokoohi

Academic Editor

PLOS ONE

Journal Requirements:

https://www.frontiersin.org/journals/cellular-and-infection-microbiology/articles/10.3389/fcimb.2022.1027424/full

In your revision ensure you cite all your sources (including your own works), and quote or rephrase any duplicated text outside the methods section. Further consideration is dependent on these concerns being addressed.

4. Thank you for stating the following financial disclosure: "SM: JST SPRING Grant Number JPMJSP2111

TS: JSPS KAKENHI Grant Number 22K07054, MW: JSPS KAKENHI Grant Number 21H02360".

5. Thank you for stating the following in the Acknowledgments Section of your manuscript: "This work was supported by JST SPRING Grant Number JPMJSP2111, JSPS KAKENHI Grant Number 22K07054, JSPS KAKENHI Grant Number 21H02360, and Molecular Profiling Committee, Grant-in-Aid for Transformative Research Areas “Advanced Animal Model Support (AdAMS)” from JSPS KAKENHI Grant Number JP 22H04922."

Please remove any funding-related text from the manuscript and let us know how you would like to update your Funding Statement. Currently, your Funding Statement reads as follows: "SM: JST SPRING Grant Number JPMJSP2111

TS: JSPS KAKENHI Grant Number 22K07054, MW: JSPS KAKENHI Grant Number 21H02360".

Additional Editor Comments:

Dear Authors

We have received the feedback for your work. You need to address the concerns of the Referees. The comments are given for you reference. Please note that I have acted as a reviewer for this manuscript, and you will find my comments below, under Reviewer 1.

Reviewers' comments:

Reviewer's Responses to Questions

**Comments to the Author**

1. Is the manuscript technically sound, and do the data support the conclusions?

Reviewer #1: Yes

Reviewer #2: Yes

2. Has the statistical analysis been performed appropriately and rigorously? 

Reviewer #1: Yes

Reviewer #2: Yes

3. Have the authors made all data underlying the findings in their manuscript fully available?

Reviewer #1: Yes

Reviewer #2: Yes

4. Is the manuscript presented in an intelligible fashion and written in standard English?

Reviewer #1: Yes

Reviewer #2: Yes

5. Review Comments to the Author

Reviewer #1: The paper entitled Effect of the JAK2/STAT3 Pathway on Internalization of Francisella novicida; is very interning. It was written well and provided with sufficient data. However, some concern need to be addressed.

1-Why the authors choose Cucurbitacin I? What about the other inhibitors?

2-What is the source of Cucurbitacin I?

3-What is the purpose of providing merge, DIC ,and GFP?

4-Interfiering of the JAK2/STAT3 Pathway by Cucurbitacin, how it affect the phenotype of the bacteria?

5-The title of the paper is not in line with the context. It should be revised by the authors. Internalization, and withing the text is all about inhibiting the pathway by Cucurbitacin I.

Reviewer #2: The paper was checked as Effect of the JAK2/STAT3 Pathway on Internalization of Francisella novicida

It is an exciting paper on the plant extract main ingredient effect on the bacteria, which bring valuable insight into the bacterial physiology and biology within the cells. However, the authors must explain how they choose Cucurbitacin I, and what was the rate of inhibition? The title of the paper also confusing as only bacteria and the pathway included, but not the effect of plant extract on the pathway. Overall, the manuscript is well written, and I would recommend to be published in PLoS One after a minor revision.

6. PLOS authors have the option to publish the peer review history of their article (what does this mean?). If published, this will include your full peer review and any attached files.

Reviewer #1: No

Reviewer #2: No

---

## [Author Response · Author response to Decision Letter 0]

21 Aug 2024

Thank you for reviewing our paper entitled '' Effect of the JAK2/STAT3 Pathway on Internalization of Francisella novicida'' by Matsumoto et al. (PONE-D-24-27526), and giving us an opportunity to submit our manuscript again. 

We corrected our manuscript according to comments of the editor and reviewers. We believe that the revised manuscript is improved significantly with the referees’ advice and now acceptable for publication in PLOS One.

1. When submitting your revision, we need you to address these additional requirements. Please ensure that your manuscript meets PLOS ONE's style requirements, including those for file naming. The PLOS ONE style templates can be found at 

--As you suggested, we checked the requirements and made small changes, e.g. to font sizes.

https://www.frontiersin.org/journals/cellular-and-infection-microbiology/articles/10.3389/fcimb.2022.1027424/full

In your revision ensure you cite all your sources (including your own works), and quote or rephrase any duplicated text outside the methods section. Further consideration is dependent on these concerns being addressed.

-- As you suggest, we checked our manuscript using iThenticate software and rephrase some sentences (Line 20-21, 91-92, 161-162).

-- As you suggest, we arranged the ‘Funding Information’.

4. Thank you for stating the following financial disclosure: "SM: JST SPRING Grant Number JPMJSP2111 TS: JSPS KAKENHI Grant Number 22K07054, MW: JSPS KAKENHI Grant Number 21H02360".

--As you suggested, we added the sentence in cover letter.

5. Thank you for stating the following in the Acknowledgments Section of your manuscript: "This work was supported by JST SPRING Grant Number JPMJSP2111, JSPS KAKENHI Grant Number 22K07054, JSPS KAKENHI Grant Number 21H02360, and Molecular Profiling Committee, Grant-in-Aid for Transformative Research Areas “Advanced Animal Model Support (AdAMS)” from JSPS KAKENHI Grant Number JP 22H04922."

Please remove any funding-related text from the manuscript and let us know how you would like to update your Funding Statement. Currently, your Funding Statement reads as follows: "SM: JST SPRING Grant Number JPMJSP2111

TS: JSPS KAKENHI Grant Number 22K07054, MW: JSPS KAKENHI Grant Number 21H02360".

--Thank you for checking our funding information. We removed Acknowledge section from the manuscript. The information listed above is correct. We added explanation about this information in cover letter.

--We checked references and found no problems.

Reviewers' comments:

Reviewer's Responses to Questions

Comments to the Author

Reviewer #1: The paper entitled Effect of the JAK2/STAT3 Pathway on Internalization of Francisella novicida; is very interning. It was written well and provided with sufficient data. However, some concern need to be addressed.

--Thank you for your reviewing. As you suggested we corrected our manuscript.

1- Why the authors choose Cucurbitacin I? What about the other inhibitors?

--In this study, we identified 56 inhibitors that negatively regulate F. novicida infection. Among them 3 inhibitors were related to Jak-2/STAT3 pathway. Therefore we thought that Jak-2/STAT3 pathway might be involved in F. novicida infection. Cucurbitacin was used as a representative inhibitor of the Jak-2/Stat3 pathway. Additionally, other promising inhibitors have been identified, including those targeting mTOR and p53, which are currently undergoing analysis. To explain this we rewrote sentences (line 197-201).

2- What is the source of Cucurbitacin I?

--Cucurbitacin I is a natural cell-permeable triterpenoid isolated from Cucurbitaceae, and is a selective inhibitor of JAK2/STAT3. The Cucurbitacin I is explained in Discussion section (line334-335).

3- What is the purpose of providing merge, DIC ,and GFP?

--DIC was used to show the location of the cells; the intention was to show that the bacteria were intracellularly located by using merged images with the DIC.

4- Interfiering of the JAK2/STAT3 Pathway by Cucurbitacin, how it affect the phenotype of the bacteria?

--We thought cucurbitacin might have an antibiotic effect, so we added cucurbitacin to the medium and investigated growth, but found that cucurbitacin had no effect on bacterial growth (Fig 2). From these data we consider that cucurbitacin does not affect the phenotype of F. novicida.

5- The title of the paper is not in line with the context. It should be revised by the authors. Internalization, and withing the text is all about inhibiting the pathway by Cucurbitacin I.

--We changed the title as you suggested (line 1).

Reviewer #2: The paper was checked as Effect of the JAK2/STAT3 Pathway on Internalization of Francisella novicida. It is an exciting paper on the plant extract main ingredient effect on the bacteria, which bring valuable insight into the bacterial physiology and biology within the cells.

--Thank you for your review. We corrected our manuscript as you suggested.

However, the authors must explain how they choose Cucurbitacin I, and what was the rate of inhibition?

--In this study, we identified 56 inhibitors that negatively regulate F. novicida infection. Among them 3 inhibitors were related to Jak-2/STAT3 pathway. Therefore we thought that Jak-2/STAT3 pathway might be involved in F. novicida infection. Cucurbitacin was used as a representative inhibitor of the Jak-2/Stat3 pathway. Additionally, other promising inhibitors have been identified, including those targeting mTOR and p53, which are currently undergoing analysis. To explain this we rewrote sentences (line 197-201). In the first screening we used the intensity of GFP, so the rate of inhibition could only be expressed as a relative value compared to the control (table S1). Therefor, we measured the effect of Cucurbitacin I on F. novicida infection in Fig. 1.

The title of the paper also confusing as only bacteria and the pathway included, but not the effect of plant extract on the pathway.

--We changed the title as you suggest(line 1).

Overall, the manuscript is well written, and I would recommend to be published in PLoS One after a minor revision.

--We checked figures with PACE and converted tiff files were uploaded to the journal online system.

---

## [Editor Report · Decision Letter 1]

26 Aug 2024

Role of the JAK2/STAT3 Pathway on Infection of Francisella novicida.

PONE-D-24-27526R1

Dear Dr.Takashi Shimizu,

We’re pleased to inform you that your manuscript has been judged scientifically suitable for publication and will be formally accepted for publication once it meets all outstanding technical requirements.

Kind regards,

Ebrahim Shokoohi

Academic Editor

PLOS ONE

Additional Editor Comments (optional):

Authors improved the paper and answered all raised concerns by the Referees.

Reviewers' comments:

no comments

---

## [Editor Report · Acceptance letter]

30 Aug 2024

PONE-D-24-27526R1 

PLOS ONE

Dear Dr. Shimizu, 

I'm pleased to inform you that your manuscript has been deemed suitable for publication in PLOS ONE. Congratulations! Your manuscript is now being handed over to our production team.

Kind regards, 

on behalf of

Dr. Ebrahim Shokoohi 

Academic Editor

PLOS ONE